# Genomic Mapping of Epidermal Growth Factor Receptor and Mesenchymal–Epithelial Transition-Up-Regulated Tumors Identifies Novel Therapeutic Opportunities

**DOI:** 10.3390/cancers15123250

**Published:** 2023-06-20

**Authors:** Lucía Paniagua-Herranz, Bernard Doger, Cristina Díaz-Tejeiro, Adrián Sanvicente, Cristina Nieto-Jiménez, Víctor Moreno, Pedro Pérez Segura, Balazs Gyorffy, Emiliano Calvo, Alberto Ocana

**Affiliations:** 1Experimental Therapeutics in Cancer Unit, Medical Oncology Department, Hospital Clínico San Carlos (HCSC), Instituto de Investigación Sanitaria (IdISSC) and CIBERONC, 28040 Madrid, Spain; lucia.paniagua@salud.madrid.org (L.P.-H.); cristina.diaztejeiro@salud.madrid.org (C.D.-T.); adrian.sanvicente@salud.madrid.org (A.S.); cnietoj@salud.madrid.org (C.N.-J.); pedro.perez@salud.madrid.org (P.P.S.); 2START Madrid-HM Centro Integral Oncológico Clara Campal (CIOCC), Early Phase Program, HM Sanchinarro University Hospital, 28050 Madrid, Spain; bernard.doger@startmadrid.com (B.D.); emiliano.calvo@startmadrid.com (E.C.); 3Facultad de Ciencias Químicas, Universidad Complutense de Madrid, 28040 Madrid, Spain; 4START Madrid-Fundación Jiménez Díaz (FJD), Early Phase Program, Fundación Jiménez Díaz Hospital, 28040 Madrid, Spain; victor.moreno@startmadrid.com; 5Department of Bioinformatics, Semmelweis University, 1094 Budapest, Hungary; gyorffy.balazs@yahoo.com; 6Department of Pediatrics, Semmelweis University, 1094 Budapest, Hungary; 7TTK Cancer Biomarker Research Group, Institute of Enzymology, 1117 Budapest, Hungary

**Keywords:** EGFR, MET, bi-specific antibodies, anti-PD-L1, pancreatic adenocarcinoma, prostate adenocarcinoma, immune checkpoint therapies

## Abstract

**Simple Summary:**

The identification of proteins in the cellular membrane of the tumoral cell is a key to the design and guidance of therapeutic agents. Recently, a bi-specific antibody termed amivantamab, targeting epidermal growth factor receptor (EGFR) and mesenchymal–epithelial transition factor (MET), which are oncogenic membrane proteins, received regulatory approval for the treatment of adult patients with locally advanced or metastatic non-small cell lung cancer (NSCLC). In this study, the authors explored tumor types with high levels of expression of EGFR and MET and focused on prostate adenocarcinoma and pancreatic adenocarcinoma, where anti-PD(L)1 agents alone have not shown relevant signs of activity. In addition, the authors confirmed that high expression of either receptor is associated with poor response to anti-PD(L)1 therapy, independently of the expression of PD-L1, suggesting that blocking these receptors with bi-specific EGFR/MET antibodies could augment the efficacy of anti-PD(L)1 inhibitors.

**Abstract:**

Background: The identification of proteins in the cellular membrane of the tumoral cell is a key to the design of therapeutic agents. Recently, the bi-specific antibody amivantamab, targeting the oncogenic membrane proteins EGFR and MET, received regulatory approval for the treatment of adult patients with locally advanced or metastatic NSCLC. Methods: The authors interrogated several publicly available genomic datasets to evaluate the expression of both receptors and PD-L1 in most of the solid and hematologic malignancies and focused on prostate adenocarcinoma (PRAD) and pancreatic adenocarcinoma (PAAD). Results: In PAAD, EGFR highly correlated with PD-L1 and MET, and MET showed a moderate correlation with PD-L1, while in PRAD, EGFR, MET and PD-L1 showed a strong correlation. In addition, in tumors treated with immune checkpoint inhibitors, including anti-PD(L)1 and anti-CTLA4, a high expression of EGFR and MET predicted detrimental survival. When exploring the relationship of immune populations with these receptors, the authors observed that in PAAD and PRAD, EGFR moderately correlated with CD8+ T cells. Furthermore, EGFR and MET correlated with neutrophils in PRAD. Conclusions: The authors identified tumor types where EGFR and MET were highly expressed and correlated with a high expression of PD-L1, opening the door for the future combination of bi-specific EGFR/MET antibodies with anti-PD(L)1 inhibitors.

## 1. Introduction

The identification of genomic alterations is a main task in cancer research [1]. Proteins expressed on the cell surface that are associated with kinase activity can be targeted either by kinase inhibitors or antibodies [1,2]. The action of membrane proteins, like HER2 and epidermal growth factor receptor (EGFR), with antibodies has demonstrated activity by inhibiting the oncogenic role of those proteins [3,4,5,6]. Examples include the anti-HER2 antibody trastuzumab or the anti-EGFR antibody cetuximab. Both have shown clinical activity in several solid tumors, like HER2-positive breast, colorectal cancer or head and neck cancer, among others [4,6,7]. In addition to the effect produced by the down-regulation of the receptor, these antibodies can stimulate the immune system through an induction of antibody-dependent cellular cytotoxicity (ADCC) mediated by the Fc fraction of the antibody [8]. On the other hand, kinase inhibitors against the kinase portion of the membrane receptor include, among others, osimertinib against EGFR or tucatinib against HER2 [9,10]. The action on membrane receptors with antibodies displays several potential benefits, including higher target specificity, therefore, avoiding undesired toxicities observed with chemical entities due to off-target effects [11,12].

More recently, novel antibody formats have been developed with the aim to improve clinical efficiency. Bi-specific antibodies were designed to bind two proteins through the Fab fraction, therefore, acting on both targets at the same time [11,13]. This approach has several potential benefits, including the fact that the antibody would be effective primarily in tumors when both proteins are present.

In solid tumors, only one antibody, amivantamab, that targets EGFR and MET, has received regulatory approval for the treatment of adult patients with locally advanced or metastatic non-small cell lung cancer (NSCLC) with EGFR exon 20 insertion mutations [14,15]. Other bi-specific agents against the same targets, like MCLA 129, are in clinical development at this moment [16]. Similarly, other bi-specific antibodies targeting different tumor-associated antigens (TAAs) in solid tumors are also in preclinical evaluation [8,11].

A further step to boost the efficacy of this family of agents is to exploit combinations of these compounds with antibodies that act on the immune response [8]. Immune checkpoint inhibitors (ICIs), like anti-PD(L)1 or anti-CTLA4 antibodies, could enhance the efficacy of bi-specific EGFR/MET antibodies by boosting T-cell activity in addition to ADCC and complement dependent cytotoxicity (CDC) [17,18].

Here, the authors interrogated genomic data to identify indications where both EGFR and MET receptors were up-regulated and, therefore, could be susceptible to be targetable with antibodies. In addition, the authors explored indications where these antibodies could be combined with immunotherapies, through the evaluation of the presence of immunomodulatory molecules or PD-L1 positivity within the selected tumors.

## 2. Materials and Methods

### 2.1. Gene Expression and Genomic Alteration Analysis

RNA-sequencing expression data to evaluate the transcriptomic levels of EGFR and MET in tumor and normal tissue were obtained from the GTEx (Genotype-Tissue Expression) and TCGA (The Cancer Genome Atlas) databases [19], using the bio informatics’ tool Gene Expression Profiling Interactive Analysis (http://gepia2.cancer-pku.cn/#index; last accessed on 6 February 2023) [20]. TCGA data contained at cBioportal [21,22] (https://www.cbioportal.org/; last accessed on 20 February 2023) [23] were considered to evaluate the amplification of EGFR and MET genes in patients with different tumors. For some indications, authors used additional sources including for: PRAD (MSK, [24]), CHOL (ICGC, [25]), PAAD (QCMG, [26]), SARC (MSK, [27]) and LUAD (MSK, [28]).

### 2.2. Correlation between Gene Expression and Immune Cell Infiltration

Tumor Immune Estimation Resource (TIMER 2.0) web server [29] (https://timer.cistrome.org/; last accessed on 8 March 2023) [30] was employed to investigate the co-expression pattern of EGFR, MET and PD-L1 across TGCA cancer types through the Gen_Corr Module. We assessed the association of EGFR and MET with immune checkpoints and tumor-infiltrating immune cells, such as CD8+ and CD4+ T cells, B cells and neutrophils, among others, across different cancer types. TIMER2.0 uses six state-of-the-art algorithms to acquire a better estimation of immune infiltration and includes 10,897 samples over 32 cancer types [31,32]. The purity and the infiltration level of immune cells were represented with a log2 TPM scale and illustrated through the “Immune Association” module of TIMER2.0. The Rho and *p*-values were obtained from Spearman’s correlation test.

### 2.3. Outcome and Prognosis Analysis

The Kaplan–Meier Plotter Online Tool [33] (https://kmplot.com/analysis/; last accessed on 22 March 2023) [34] was utilized to determine the prognostic value in solid tumors, in relation to the expression of a particular gene. The authors investigated the relationship between clinical outcome and EGFR and MET expression levels in cancer patients treated with anti-PD1 or anti-PD-L1. Overall survival (OS) was utilized as endpoint for outcome analysis. OS time was defined as the time from diagnosis to patient death or last follow-up. The KM plots are represented with the longrank *p*-value (*p*), the hazard ratio (HR) and the false-discovery rate (FDR). Genes with an HR < 1 and *p* < 0.05 predict a favorable clinical outcome.

## 3. Results

### 3.1. Expression of EGFR and MET in Cancer

The authors first evaluated the expression of EGFR and MET across a wide number of solid tumors using publicly available human genomic datasets, as described in the Materials and Methods section. EGFR was highly expressed with ≥32 transcripts per million (TPM) in glioblastoma (GBM), low-grade glioma (LGG), kidney renal clear cell carcinoma (KIRC), esophageal carcinoma (ESCA) and head and neck squamous cell carcinoma (HNSC) (Figure 1A). MET was highly expressed (≥32 TPM) in thyroid carcinoma (THCA), kidney renal papillary cell carcinoma (KIRP), KIRC and kidney chromophobe (KICH) (Figure 1B).

Indications with the highest fold change between tumor and non-transformed tissue for EGFR included thymoma (THYM) (>600-fold change), GBM (>20-fold change) and LGG (≥5-fold change) (Figure 1C,E), followed by KIRC, stomach adenocarcinoma (STAD), lung squamous cell carcinoma (LUSC), ESCA, pancreatic adenocarcinoma (PAAD), HNSC and sarcoma (SARC) (all ≥1.5-fold change) (Figure 1C). MET expression in THYM and diffuse large B-cell lymphoma (DLBC) comprised > 100-fold change (Figure 1E). ESCA, PAAD, rectum adenocarcinoma (READ), THCA, STAD, ovarian serous cystadenocarcinoma (OV), colon adenocarcinoma (COAD), testicular germ cell tumors (TGCT) and KIRP displayed a ≥5-fold change. Tumors with a ≥1.5-fold change are described in Figure 1D. Amplification of EGFR in more than 10% was observed in GBM and ESCA, and for MET, in more than 5% in OV (Figure 2A,B).

### 3.2. Co-Occurrent Expression of EGFR and MET with PD-L1 in Solid Tumors

The authors next explored the correlation between the expression of EGFR and MET individually with the expression of PD-L1 to identify indications where combinations of both therapeutic strategies could be of interest. As can be seen in Figure 3A, a positive correlation was observed for EGFR and PD-L1 in THYM, PRAD, diffuse large B-cell lymphoma (DLBC) and PAAD (Rho > 0.6), followed by THCA (Rho > 0.5) and breast-invasive carcinoma, luminal A (BRCA LumA), cholangiocarcinoma (CHOL), liver hepatocellular carcinoma (LIHC), KIRC and BRCA-basal (Rho > 0.4). For MET and PD-L1, a strong correlation was observed for uveal melanoma (UVM) (Rho > 0.6), followed by THCA, PRAD and adrenocortical carcinoma (ACC) (Rho > 0.5), as well as BLCA, THYM and lung adenocarcinoma (LUAD) (Rho > 0.4) (Figure 3B). Indications where anti-PD1 or anti-PD-L1 are approved by the US Food and Drug Administration (FDA) are marked with an arrow. Tumor types where anti-PD-(L)1 is approved was very limited and showed a modest correlation (Rho > 0.4 and ≤0.5), including for EGFR: CHOL, LIHC, KIRC and BRCA-basal, and for MET: bladder cancer (BLCA) and LUAD (Figure 3A,B). Among tumors where ICIs have not been approved, a strong correlation for both receptors was observed in PRAD and THCA, as well as a moderate correlation in PAAD and THYM.

### 3.3. Association of EGFR and MET Expression with Clinical Outcome in PD(L)1-Treated Population

The authors aimed to explore if the expression of EGFR or MET could predict outcome in patients treated with ICIs. To do so, the authors used an extensive dataset that is described in detail in Appendix A and in Section 2. As shown in Figure 4A, a high expression of EGFR and MET predicted detrimental survival in the whole population of patients treated with ICIs, including both agents anti-PD(L)1 and anti-CTLA4 (EGFR: HR = 1.76 CI = 1.48–2.09, *p* = 7.4 × 10^−11^; FDR = 1% and MET: HR = 1.42 CI = 1.2–1.69, *p* = 4.5 × 10^−5^; FDR = 2%). When selecting patients treated only with anti-PD1, the authors also observed a detrimental outcome: HR = 1.76 CI = 1.33–2.32, *p* = 5.9 × 10^−5^; FDR = 1% and HR = 1.36 CI = 1.02–1.82, *p* = 0.035; FDR 50% for EGFR and MET, respectively (Figure 4B). A similar unfavorable prognosis was identified in patients treated with anti-PD-L1 alone, if they harbored a high expression of EGFR and MET: HR = 1.25 CI = 0.99–1.58, *p* = 0.062; FDR = 100% and HR = 1.42 CI = 1.08–1.88, *p* = 0.011; FDR = over 50%, respectively (Figure 4C). Unfortunately, an analysis in some specific indications could not be performed due to the limited number of patients in those cohorts (Appendix A). However, globally, these data demonstrate that a high expression of either receptor is associated with a poor response to anti-PD(L)1 therapy, independently of the expression of PD-L1, suggesting that blocking these receptors could augment the efficacy of ICIs.

### 3.4. Co-Occurrent Expression of EGFR and MET in Solid Tumors

Next, the authors explored the correlation between EGFR and MET in all of the evaluated tumors. The authors’ aim was to identify indications where both receptors were highly present to facilitate the delivery of the antibody to the tumor. EGFR and MET strongly correlated in THYM (Rho = 0.759, *p* = 7.76 × 10^−23^), BRCA-LumA (Rho = 0.754, *p* = 2.62 × 10^−96^), PAAD (Rho = 0.699, *p* = 2.31 × 10^−26^), BRCA (Rho = 0.686, *p* = 2.25 × 10^−139^), THCA (Rho = 0.675, *p* = 3.33 × 10^−66^), HNSC-HPV+ (Rho = 0.672, *p* = 5.78 × 10^−13^), KIRP (Rho = 0.633, *p* = 3.02 × 10^−30^); and with less strength PRAD (Rho = 0.599, *p* = 8.48 × 10^−42^), HNSC (Rho = 0.588, *p* = 3.99 × 10^−47^), HNSC-HPV- (Rho = 0.566, *p* = 2.48 × 10^−35^), READ (Rho = 0.565, *p* = 4.12 × 10^−13^), BLCA (Rho = 0.548, *p* = 2.67 × 10^−30^), KICH (Rho = 0.531, *p* = 5.37 × 10^−6^), MESO (Rho = 0.527, *p* = 2.16 × 10^−7^), KIRC (Rho = 0.518, *p* = 5.17 × 10^−33^) and BRCA-basal (Rho = 0.513, *p* = 4.35 × 10^−13^) (Figure 5A).

### 3.5. Therapeutic Opportunities to Exploit Anti-EGFR/MET with Anti-PD-(L)1 Antibodies

Taking into consideration the data described before, the authors observed some potential opportunities for clinical development in indications where anti-PD(L)1 agents alone have not shown relevant signs of activity. In this context, we selected the tumor types that also showed a moderate or high correlation between EGFR, MET and PD-L1. Among them, PAAD displayed a high correlation of EGFR and PD-L1 (Rho = 0.619, *p* = 1.79 × 10^−19^) and a moderate correlation between MET and PD-L1 (Rho = 0.398, *p* = 6.87 × 10^−8^) (Figure 5B). In addition, in this tumor, the authors observed a high correlation between EGFR and MET (Rho = 0.698, *p* = 2.31 × 10^−26^). As in Figure 5A,B, PRAD showed an association between EGFR and PD-L1 (Rho = 0.665, *p* = 1.47 × 10^−54^), as well as between MET and PD-L1 (Rho = 0.542, *p* = 3.42 × 10^−33^) and EGFR and MET (Rho = 0.598, *p* = 8.48 × 10^−42^). We also noted that EGFR, MET and PD-L1 correlated among them in THCA and THYM, as shown in Figure 5A,B. Nevertheless, the authors selected PAAD and PRAD for further analysis. Therefore, the authors decided to exclude THCA, as no information about the different histological and genomic subtypes was available (papillar, follicular, anaplastic, etc.) for a specific and detailed characterization and THYM due to the fact that this tumor originates from a lymphoid gland involved in the development of the immune system.

### 3.6. Expression of Immune Populations and Co-Inhibitors in PAAD and PRAD

Next, the authors analyzed the association of EGFR and MET expression with immune infiltration levels in PAAD and PRAD. For PAAD, a positive but moderate correlation was seen between EGFR and CD8+ T cells (Rho = 0.403, *p* = 4.57 × 10^−8^), neutrophils (Rho = 0.408, *p* = 2.92 × 10^−8^) and myeloid dendritic cells (DCs) (Rho = 0.426, *p* = 6.50 × 10^−9^) (Figure 6A,B). No association with CD4+ T cells, B cells and macrophages was observed. On the other hand, the presence of MET did not correlate with any of the evaluated immune cells (Figure 6A,B). The authors also observed a positive correlation in PRAD between EGFR and CD8+ T cells (Rho = 0.457, *p* = 8.19 × 10^−23^) (Figure 6A). In addition, EGFR and MET correlated both with neutrophils (Rho = 0.538, *p* = 1.59 × 10^−23^ and Rho = 0.638, *p* = 5.94 × 10^−49^, respectively) and myeloid dendritic cells (Rho = 0.449, *p* = 4.48 × 10^−22^ and Rho = 0.534, *p* = 5.53 × 10^−32^, respectively) (Figure 6B). In this tumor, MET also had a positive correlation with CD4+ T cells (Rho = 0.438, *p* = 6.42 × 10^−21^) (Figure 6A). No correlation was found for EGFR and MET with CD4+ T cells and macrophages (Figure 6A,B).

Apart from the correlation observed between both receptors and PD-L1, the authors analyzed the association with co-inhibitor immune checkpoint inhibitors, detecting only a moderate correlation (Rho = 0.460, *p* = 3.21 × 10^−23^) between MET and HAVCR2 in PRAD (Figure 6C).

## 4. Discussion

In this article, the authors mapped the transcriptomic expression of EGFR and MET in solid tumors and their association with PD-L1 expression. The authors’ goal was to identify indications where combinations of bi-specific antibodies against EGFR/MET could be utilized in combination with anti-PD-L1 antibodies.

First, the authors identified indications where EGFR and MET were highly and differentially expressed compared to normal tissues, including for EGFR: THYM, GBM and LGG, and for MET: THYM and DLBC. On the other hand, gene amplification was not significantly observed, apart from GBM and ESCA for EGFR (more than 10%) and OV for MET (more than 5%).

When exploring the contribution to the anti-PD (L) 1 response of EGFR and MET receptors, the authors confirmed a clear detrimental effect. A high expression of EGFR and MET in a dataset of patients treated with ICIs was associated with poor survival. These data verified the authors’ previous observations describing the lack of activity of anti-PD (L) 1 antibodies in lung cancer patients with activation of EGFR due to kinase mutations [35,36,37,38]. Little information is available regarding the contribution of MET alterations, mainly MET exon 14 skipping mutations, in relation to the response to anti-PD(L)1 therapies. Data in this regard are restricted to a very small number of patients [39,40]. Of note, for other membrane tirosine kinase (TK) receptors, including ErbB2, activation of their pathway has been associated with a lack of response to ICIs [41,42]. In line with this, tumors with alterations in the ErbB family genes (and particularly EGFR) develop a non-inflamed tumor microenvironment, which could be due to a low tumor mutational burden (TMB) [43]. In our study, it is unknown if tumors with a high presence of EGFR and MET associate with low TMB.

A clear example that was not evaluated previously includes PAAD, where EGFR and MET correlated with each other and with PD-L1. This tumor also had a mild correlation between EGFR and CD8+ T cells and dendritic cells. Likewise, in PRAD, EGFR and MET correlated with the presence of PD-L1, and in addition to that, EGFR had a mild correlation with CD8+ T cells, neutrophils and DCs, while MET correlated strongly with neutrophils.

In these two indications, PAAD and PRAD, the activity of ICI was limited. For instance, in PRAD, anti-PD1 antibodies have shown a lack of efficacy in monotherapy or in combination with hormonotherapy, suggesting the necessity for the evaluation in combination with other immune-modulating agents [44,45]. In addition, in this indication, a relevant role for ErbB receptors and ligands has been reported in different studies, suggesting potential use as a therapeutic target [46]. In line with this, classical MET expression has been associated with androgen-resistant prostate cancer tumors [47]. In PAAD, therapeutic actions against EGFR either with kinase inhibitors or with antibodies have shown disappointing results [48], and it is known that this tumor harbors an immune-suppressive environment [49]. Very recently published preclinical data suggest augmented antitumoral activity using in vivo models of the dual targeting of EGFR and MET in this particular tumor type [50].

A combination of antibodies with anti-PD-(L)1 inhibitors can potentiate the activity of each agent alone through the activation of ADCC and CDC immunologic mechanisms [8]. Antibodies with FC modifications to augment ADCC and CDC activity have demonstrated clinical activity, like tafasitamab in DLBC [51]. In this context, it is expected that combinations of bi-specific antibodies against EGFR/MET would enhance the effect when combined with anti-PD-(L)1 agents. Finally, examples of bi-specific PD-L1 and EGFR antibodies are currently in preclinical development [52]. Similar examples have been reported for MET and PD1 bi-specific antibodies [53].

The authors’ study has limitations. First, this is a bioinformatic analysis using publicly available genomic datasets. Therefore, the authors may expect that in the near future, the findings described here could be confirmed with immunohistochemistry studies using human samples. Secondly, the combinations suggested here should also be evaluated preclinically. In addition, we acknowledge that genetic variations, including single-nucleotide polymorphisms (SNPs) or EGFRex20 insertions, which can lead to differences in ethnicities or therapeutic efficacies, could not be explored due to the lack of available datasets.

## 5. Conclusions

In summary, the authors have described tumor types, where the presence of EGFR/MET is associated with detrimental response to anti-PD(L)1 therapies and with an immune-suppressive environment. Therefore, the authors suggest that co-targeting EGFR/MET and PD(L)1 with antibodies could enhance treatment efficacy, particularly in those tumors with high correlation among receptors, like PAAD and PRAD.

## Figures and Tables

**Figure 1 cancers-15-03250-f001:**
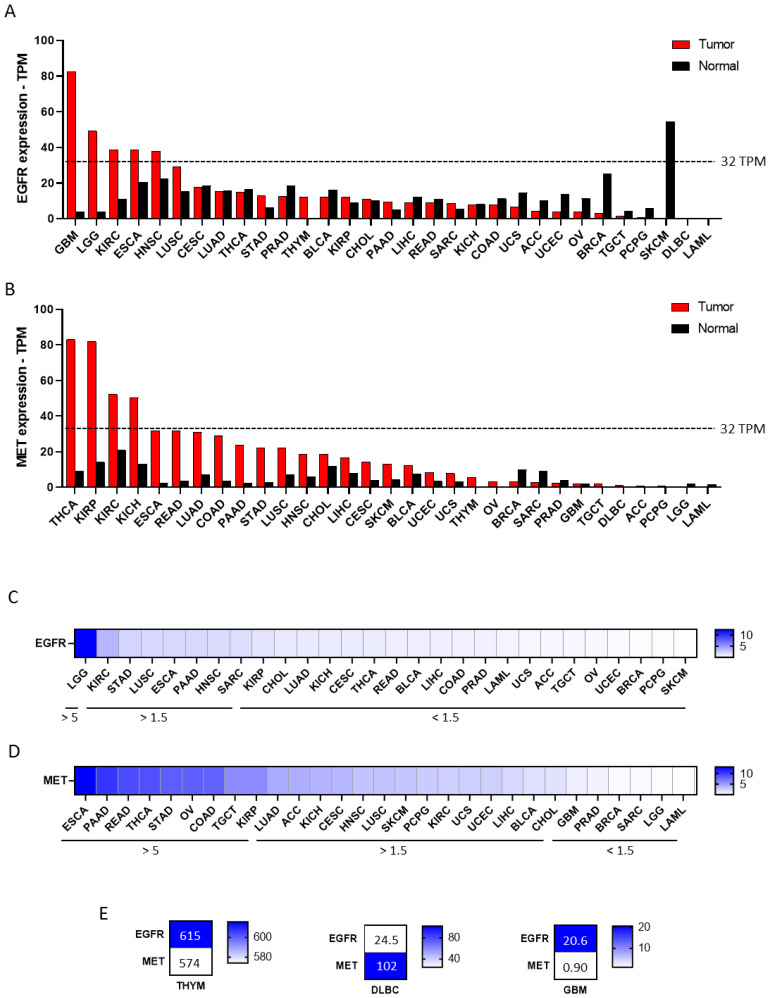
EGFR and MET expression profile across all tumor samples and paired normal tissues. (**A**) EGFR expression levels in different cancers validated using GEPIA2 database. EGFR was highly expressed (TPM > 32) in GBM, LGG, KIRC, ESCA and HNSC. (**B**) MET expression levels across different cancers validated using GEPIA2 database. MET was highly expressed (TPM > 32) in THCA, KIRP, KIRC and KIRCH. (**C**) Heat map depicting fold change between tumor and non-transformed tissue for EGFR and MET (**D**). (**E**) Heat maps displaying information from those indications with a fold change >20 between tumor and non-transformed tissue for EGFR and MET.

**Figure 2 cancers-15-03250-f002:**
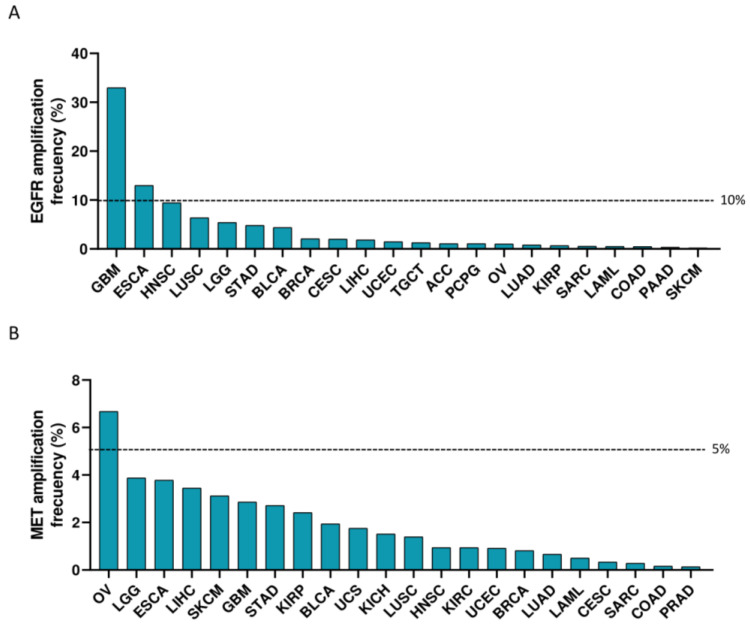
Amplification of EGFR (**A**) and MET (**B**) across tumor indications.

**Figure 3 cancers-15-03250-f003:**
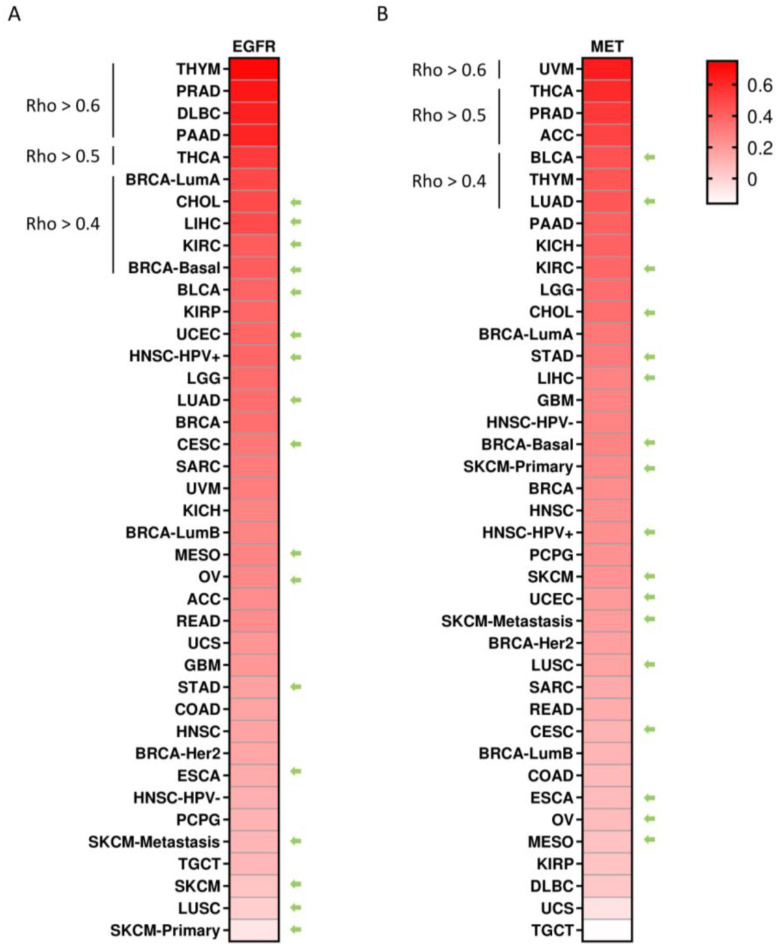
Co-occurrent expression of EGFR and MET with PD-L1 expression in tumor indications. Heat map displaying the Spearman correlation coefficient of EGFR (**A**) and MET (**B**) with PD-L1. Green arrows indicate tumors where anti PD1 or anti PD-L1 antibodies are approved by the US food and drug administration (FDA).

**Figure 4 cancers-15-03250-f004:**
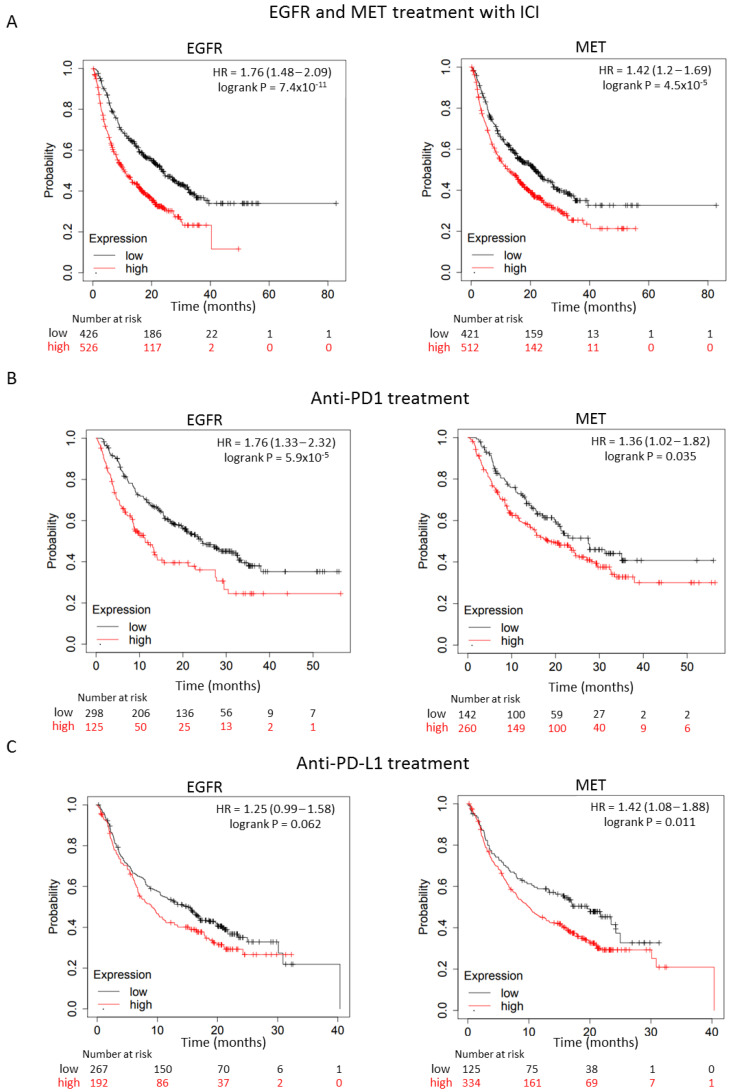
Association between EGFR or MET expression levels and overall survival (OS) in patients treated with immune check point inhibitors (ICIs). (**A**) Kaplan-Meier survival curves comparing low and high expression of EGFR (OS; n = 954) and MET (OS; n = 933) in the whole population of patients treated with ICIs, including both anti-PD(L)1 and anti-CTLA4. (**B**) Kaplan-Meier survival curves comparing low and high expression of EGFR (**left panel**) and MET (**right panel**) in PD1-treated population (OS; n = 423 and OS; n = 402, respectively) and (**C**) in PD-L1-treated population (OS; n = 459 and OS; n = 459, respectively).

**Figure 5 cancers-15-03250-f005:**
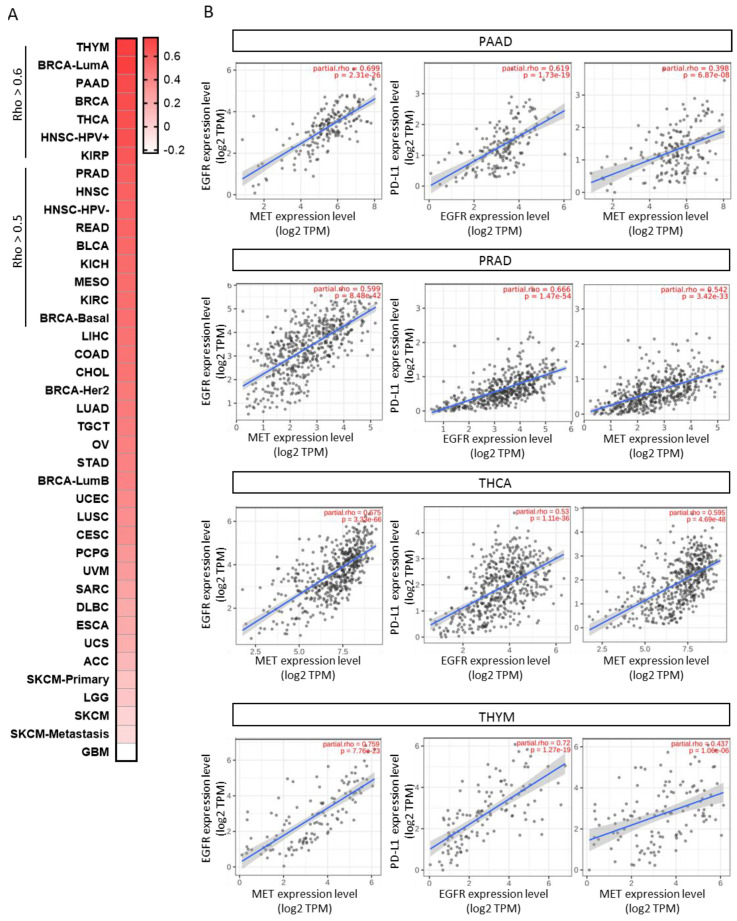
Correlation between EGFR and MET expression levels in TCGA tumors. (**A**) Heat map depicting the Spearman correlation coefficient between EGFR and MET expression levels using TIMER2.0. (**B**) Correlation between PD-L1, EGFR and MET expression levels estimated using TIMER2.0 in PAAD, PRAD, THCA and THYM.

**Figure 6 cancers-15-03250-f006:**
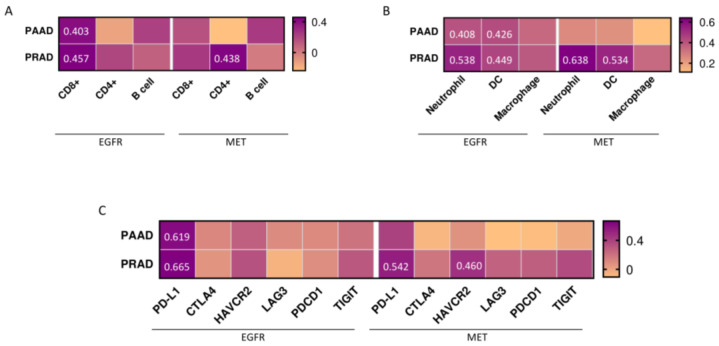
Association of EGFR and MET expression levels with immune infiltrates and co-inhibitor immune checkpoints in PAAD and PRAD. Heat map depicting the Spearman correlation coefficient between EGFR and MET expression and the presence of (**A**) CD8+ T cells, CD4+ T cells, B cells and (**B**) neutrophils, dendritic cells (DCs) and macrophages. Numbers inside the heat maps indicate the rho value. (**C**) Heat map depicting the Spearman correlation coefficient between EGFR and MET with inhibitory checkpoint molecules.

## Data Availability

All data generated or analyzed during this study are included in this published article.

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
