# Peer review of "Genomic Mapping of Epidermal Growth Factor Receptor and Mesenchymal–Epithelial Transition-Up-Regulated Tumors Identifies Novel Therapeutic Opportunities"

_cancers, 2023, doi:10.3390/cancers15123250_

Round 1

Reviewer 1 Report

v                 Suggestions to Author/s

1. Dear Dr. Alberto Ocana, as a selected reviewer, I made the prompt check of your excellent scientific article: “Genomic mapping of EGFR and MET up-regulated tumors identify novel therapeutic opportunities” and found it: (x) Excellent, accept the submission (5)

2. To make the prompt check, your article in pdf was converted into doc. The text conversion was made by the “Convertio online”.

3. In your text we.. was changed into like: authors suggestions…

4. In the text some small mistakes were found and corrected. They are highlighted in the red color. Please accept the corrections and highlighted back into black color.

5. You are kindly asked to give your text to the lector for final editing and language.

5. After careful check of your corrected text, please use the “Convertio online” and convert back from doc to pdf. I sincerely hope that the form of so obtained pdf will not be different from original pdf of your article.

Authors are kindly asked to give theirs' text to the lector for final editing and language.

Author Response

Respond to reviewer

We have modified the manuscript in accordance with the suggestions given by

Reviewer 2 Report

This is an interesting study that evaluates the “Genomic mapping of EGFR and MET upregulated tumors 2 identify novel therapeutic opportunities.”. However, the manuscript can be improved with some points below that the authors should consider.

1.In the manuscript, abbreviations are fully written out upon their first occurrence. For example, in L19: EGFR (epidermal growth factor receptor) and MET (mesenchymal-epithelial transition factor). This helps readers understand them easily.

2.In Figure 1, Panel C only includes 28 types of tumors. Could the data for GBM (glioblastoma) be missing? Please double-check.

3.The font in Figure 4 and Figure 5B is not very clear. Additionally, in Fig 4A, the author can add a title, such as "(EGFR and MET treatment with ICI)".

4.The author evaluated the expression of EGFR/MET in various solid tumors using human genomic data and focused on the expression of PD-L1 in PRAD and PAAD. Are there any reports in the clinical setting utilizing bi-specific antibodies to improve the efficacy of treatment?

5.Due to human genetic variations leading to single nucleotide polymorphisms (SNPs), has the study considered different ethnicities when analyzing the transcriptomic expression of EGFR and MET in solid tumors, which may result in different outcomes?

6.The authors suggest that using EGFR/MET and PD(L)1 could enhance the therapeutic efficacy for tumors with a high correlation between EGFR and MET. Have they considered EGFRex20ins+ patients or unselected patients in their recommendation?

Author Response

Respond to reviewer

----------------------------------

This is an interesting study that evaluates the “Genomic mapping of EGFR and MET upregulated tumors 2 identify novel therapeutic opportunities.”. However, the manuscript can be improved with some points below that the authors should consider.

Point 1.In the manuscript, abbreviations are fully written out upon their first occurrence. For example, in L19: EGFR (epidermal growth factor receptor) and MET (mesenchymal-epithelial transition factor). This helps readers understand them easily.

Response: We have modified this accordingly. Please see the attachment.

Point 2.In Figure 1, Panel C only includes 28 types of tumors. Could the data for GBM (glioblastoma) be missing? Please double-check.

Response: The tumors where the fold change between tumor and non-transformed tissue is above 20 (GBM, THYM and DLBC) are shown in Figure 1, Panel E

Point 3.The font in Figure 4 and Figure 5B is not very clear. Additionally, in Fig 4A, the author can add a title, such as "(EGFR and MET treatment with ICI)".

Response: This has been changed in the manuscript.

Point 4.The author evaluated the expression of EGFR/MET in various solid tumors using human genomic data and focused on the expression of PD-L1 in PRAD and PAAD. Are there any reports in the clinical setting utilizing bi-specific antibodies to improve the efficacy of treatment?

Response: we have included all available information regarding pre-clinical studies exploring EGFR/PD (L)1 or MET/PD (L)1 antibodies

Point 5.Due to human genetic variations leading to single nucleotide polymorphisms (SNPs), has the study considered different ethnicities when analyzing the transcriptomic expression of EGFR and MET in solid tumors, which may result in different outcomes?

Response: this is a very interesting observation. Unfortunately, this kind of data is not available within the current existing sources. However, we have added a sentence in the discussion section acknoweldge this concept: “In addition, we acknowledge that genetic variations including single nucleotide polymorphisms (SNPs) or EGFRex20 insertions, that can lead to differences in ethnicities or therapeutic efficacies, could not be explored due to the lack of available datasets”

Point 6.
The authors suggest that using EGFR/MET and PD(L)1 could enhance the therapeutic efficacy for tumors with a high correlation between EGFR and MET. Have they considered EGFRex20ins+ patients or unselected patients in their recommendation?

Response: again this is a relevant concept. With the current available datasets we can not extract this information. However, we have added this concept in the discussion section: In addition, we acknowledge that genetic variations including single nucleotide polymorphisms (SNPs) or EGFRex20 insertions, that can lead to differences in ethnicities or therapeutic efficacies, could not be explored due to the lack of available datasets.
